# A Feasibility Study of Two Cognitive Training Programs for Urban Community-Dwelling Older Adults

**Sara Benham** [1,*] **, Kelly Otchet** [2]**, Diana Senft** [3] **and Ann Marie Potter** [1]

1   Department of Rehabilitation Sciences, Moravian University, Bethlehem, PA 18018, USA;
    pottera@moravian.edu
2   Department of Occupational Therapy, University of the Sciences in Philadelphia,
    Philadelphia, PA 19104, USA; k.otchet@usciences.edu
3   NaviHealth, Brentwood, TN 37027, USA; dianamiller742@gmail.com
*   Correspondence: benhams@moravian.edu

**Abstract:** Cognitive training approaches are promising to manage the effects of normal cognitive decline for the aging adult, especially with the development and integration of computerized cognitive training. Supportive community models for older adults, such as senior centers, may provide engagement opportunities for occupation-based cognitive training programming. Fourteen older adults (*n* = 13 Black) from an urban older adult community center participated. This feasibility trial used a two-group, pretest-posttest design to examine differences between an occupation-based computerized cognitive training (CCT) program (*n* = 7) and a traditional cognitive training (TCT) program (*n* = 7), as assessed by participants' perceptions of the perceived benefits, tolerance of time of sessions, and on executive functioning measures. There were no significant differences in the tolerance of time of sessions (*p* = 0.81) between CCT (average session time = 43.64 min) and TCT (average session time = 44.27 min). Additionally, there were no significant differences in how the two program groups perceived the training based on helpfulness (*p* = 1.00), positive opinions (*p* = 0.46), and executive functioning measurement changes. All participants reported "enjoyment" of the training. Including occupation-based CCT and TCT programming is feasible and positive within community-based programming focusing on a diverse population. Short-term improvements in executive functioning should not be expected but are worthy of longer-term observation, considering a socialization component, telehealth integrations, and expansion of supportive technology-based models.

**Keywords:** cognitive training; older adult; senior centers; occupational therapy





## 1. Introduction

The naturally occurring physical and cognitive changes associated with aging raise concerns for maintaining independence and safely managing tasks within the home and community. Cognitive changes associated with aging include decreased abilities in alternating and divided attention, working memory, declarative memory, and executive functioning [1,2]. Successful aging may be characterized as an individual's ability to function with sustained cognitive and physical abilities, social and productive engagement, and adapt to new roles as the body structures and functions change [3]. Aging in place or living independently in one's own home or retirement community runs hand in hand with successful aging, as 94% of individuals who met the criteria of successful aging were living independently [3]. Aging in place is a widespread initiative promoted in occupational therapy (OT) service provision and other health professions, emphasizing independence within one's long-standing and personal environment through adaptation. Since 1965, US policy has supported successful aging through the Older American Act (OAA). This act provides funding for community support-based services, such as senior centers, to help older adults stay as engaged and independent as possible and prevent injury and hospitalization through supportive health services and programming [4]. Cognitive training may

be a potential programming solution to address engagement to sustain cognitive abilities. Cognitive training, defined in this study as structured intellectual activities that target specific skills which a skilled therapist oversees, may be effective in decreasing the effects of cognitive decline for older adults [5]. The positive impact of engagement in cognitive activity throughout the lifespan has been well-documented from middle age to older adulthood [6,7]. Tranter and Koutstaal's findings show that paper and pencil cognitive training for 10–12 weeks, along with other forms of cognitively stimulating activity, decreased cognitive decline in older adults [8].

Further comprehensive findings from a systematic review concluded that specific skills might improve after traditional cognitive training (TCT) for older adults, including memory, attention, and executive functioning. However, generalizations to everyday life performance are undetermined [9]. Cognitive training approaches may effectively decrease the effects of cognitive decline, and more specifically, reasoning training may promote IADL performance improvements [5,6,10]. Over time, community senior centers have altered programming to encourage a lifelong learning model focusing on intellectual stimulation for older adults, and therefore program development may be warranted [4].

Community-based OT interventions are associated with improvements in the ability to participate in work, activities, and vitality [11]. OT practitioners have a strong tradition in providing cognitive interventions with the ability to apply training programs to community-based settings, such as senior centers. OT practitioners use several cognitive training methods, including TCT paper-and-pencil tasks and computerized cognitive training (CCT). CCT is a customized training alternative that utilizes a technology-based platform to address specific cognitive skills [12]. Benefits of CCT may include increases in cognitive function of skill-based measures of global cognition, memory, language, visuospatial skills, and executive functioning [13]. Still, similar to traditional cognitive training program outcomes, functional task performance enhancements are limited [14–16]. CCT for older adults may be less labor-intensive and equally effective as paper-and-pencil traditional cognitive training (TCT) approaches [17]. Miller et al. recommend that self-pacing through CCT may be an option. However, self-pacing was not trialed through the study protocol [16]. As occupational therapists and other rehabilitation professionals consider pandemic-related social distancing options through the expansion of mediums of telecommunication remote access or home exercise recommendations [18], the client's preferences in regard to the type of programming regarding TCT compared to CCT are unknown. In addition, no studies have examined client tolerance of the activities during self-pacing and the feasibility with a racially diverse sample of older adults. Few studies have made concerted efforts to recruit non-white races, with Black participants underrepresented overall [10]. Although the evidence may suggest that dementia prevalence is higher in Black Americans, the rate of cognitive decline among Black Americans is not faster than in white counterparts. This may reflect persistent differences in access disparities, with more recent calls to combat these disparities into older age and the progressive cognitive aging stages [19,20].

This study explores the comparisons and feasibility of delivering two cognitive training programs, CCT and TCT, in an urban-based senior center among a primarily African American or Black population. We examined changes in cognition, participants' perceptions of the programming, and tolerance for participating in cognitive training through repeated measures, surveys, and tracking of the self-paced session timing. There is limited evidence on the perception and outcomes of the different types of cognitive programming in community settings serving older adults, especially those from urban and diverse backgrounds. This study aims to inform rehabilitation professionals on the acceptance of cognitive programming for diverse populations and the feasibility of providing such programming in community-based settings.

## 2. Materials and Methods

*2.1. Research Design*

This mixed-methods feasibility study utilized a basic randomized design comparing two treatments [21]. This design provides information about the benefits and differences between two potential programs that occupational therapists or trained professionals may implement in senior centers. The variables of *cognitive changes*, the *tolerance* (amount of time per session) for participating in the 480 min of training within twelve weeks, and their *perceptions* of the programming experiences were compared between the two groups. The University of the Sciences Institutional Review Board approved and agreed to provide ethical oversight of the study.

*2.2. Participants*

Participants were initially recruited from an urban senior day center through an announcement during the member town hall. The center is part of a senior service network providing a continuum of care of activities and health services. A convenience sampling method identified potential participants through the agency member list. Members lived in their own homes or within the US Department of Housing and Urban Development (HUD) Section 202 Supportive Housing for the Elderly. The list was narrowed by excluding members, through chart review, with previously recorded Mini-Mental State Exam (MMSE) scores less than 20/30, used solely to exclude moderate-to-severe dementia for recruitment efforts, leaving 185 members remaining on the list. The member list is not regularly updated; therefore, the primary investigator consulted with the center's staff occupational therapist. An additional 29 members were excluded as they had transitioned to a memory care setting, had significant vision loss, or died. A researcher attempted to randomly approach the remaining 156 members at the center to participate in screening. Inclusion criteria for this study were that the member: was at least 55 years of age with no age limit cutoff, scored at least 17/30 on the MoCA, based on recommended mild cognitive impairment severity ranges [22], and adequate visual skills assessed by the ability to read the consent forms visually. Of the 25 members who provided their informed consent and completed the screening, 5 individuals scored below the MoCA cutoff and were excluded (see Figure 1). As this study took place in a community-based setting, participants did not disclose diagnostic or medical-related information. The principal investigator, blinded to the MoCA scores, randomly assigned participants to TCT or CCT groups by code.

*2.3. Instruments*

Regarding pretest to posttest measures of the variables of *cognitive changes*, the Montreal Cognitive Assessment (MoCA) was used to assess executive function, higher-level language, and complex visuospatial processing in the participants [23]. In addition, this assessment was used as an exclusion measure which enabled the researchers to detect mild cognitive impairment with less of a ceiling effect than the MMSE. Scores range between 0 and 30 with good reliability [24]. The Executive Function Performance Test (EFPT) was used to assess executive functioning during four selected tasks essential for self-maintenance and independent living, which are cooking, telephone use, medication intake, and bill payment [25]. A standardized cueing system for the progressive need for assistance during the initiation, organization, sequencing, safety, judgment, and completion of the four tasks is the basis for scoring the EFPT. The alternate version of the EFPT was utilized on the posttest to account for a possible learning effect from the original form, with both versions found as valid and reliable [26,27]. The Trail Making Tests Parts A and B (TMT-A and TMT-B) were used to measure visual conceptual and visuomotor tracking skills [28]. Part A and Part B scoring reflect the total time in seconds to complete the tasks, with high test-retest reliability [29]. For the EFPT, TMT-A, and TMT-B, a decrease in score indicates an improved performance.

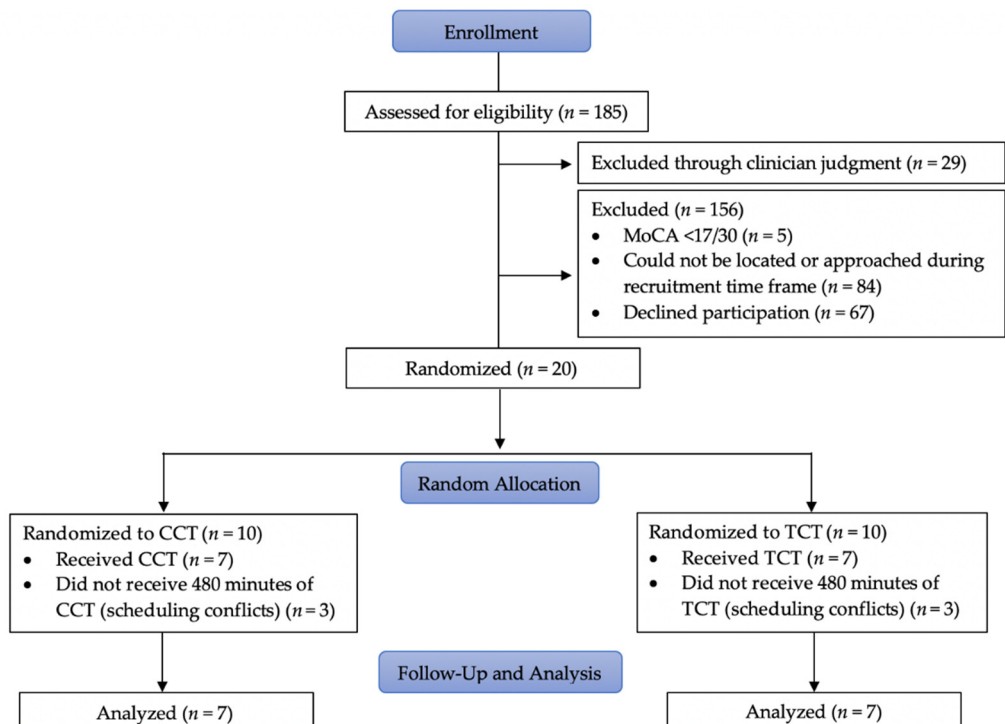

**Figure 1.** CONSORT flow diagram. Note: CCT = Computerized Cognitive Training, TCT = Traditional Cognitive Training, MoCA = Montreal Cognitive Assessment.

Researchers tracked the amount of time participating in the cognitive training activities. Upon the conclusion of the intervention phase, participants completed a six-item survey of open-ended questions providing their perceptions of the programming. The questions were designed for programmatic decision-making and reviewed by an expert in the field of qualitative research for risk of bias. The questions to gauge the aims of the programs were: 1. Do you feel that this cognitive training is helpful to you? 2. Are you enjoying the cognitive training that you are doing? 3. What is your opinion on the training sessions, and how can we help you make them better for you/for future participants? 4. Do you think you would do this cognitive training on your own while spending time at the center? 5. Do you think you would be able to set up and use the cognitive training activities without assistance? 6. Would you continue to do this cognitive training at the center after this study?

*2.4. Interventions*

Within both intervention groups, at the beginning of each session, the participants were instructed to complete the cognitive training program for a period of time duration of the participants' preference, within a range of 15 to 90 min, until they reached a total of 480 min of training within 12 weeks. Two OT student research assistants, who were directly overseen by an employed, on-site occupational therapist and unrelated to the study, monitored participants for signs of stress and cognitive fatigue throughout the training sessions for both groups and provided guidance.

The CCT group utilized the RehaCom (https://hasomed.de/en/products/rehacom/; accessed on 8 March 2022) program in the center's computer lab. RehaCom software targets specific aspects of attention, concentration, memory, perception, and problem-solving. As the training progresses, the software self-adjusts the difficulty of tasks depending on the participant's performance. Participants were exposed to a variety of occupation-based training modules of the participant's choosing, with an emphasis from the researchers to participate in real-life activity-based simulations. For example, executive function skills were required for simulated grocery shopping and planning a vacation (Figure 2).

Specific skills, such as spatial-perceptual skills, could be addressed through modules that required participants to visually estimate the size or locations of a reference image within accuracy parameters. RehaCom has been trialed with various neurological populations, including schizophrenia, multiple sclerosis, and brain injury, with positive reports on the clinical usefulness and efficacy and mixed results on the comparative effectiveness [30–32]. However, it has not been trialed with the older adult population at the study time. Research assistants trained in RehaCom supervised each participant's session.

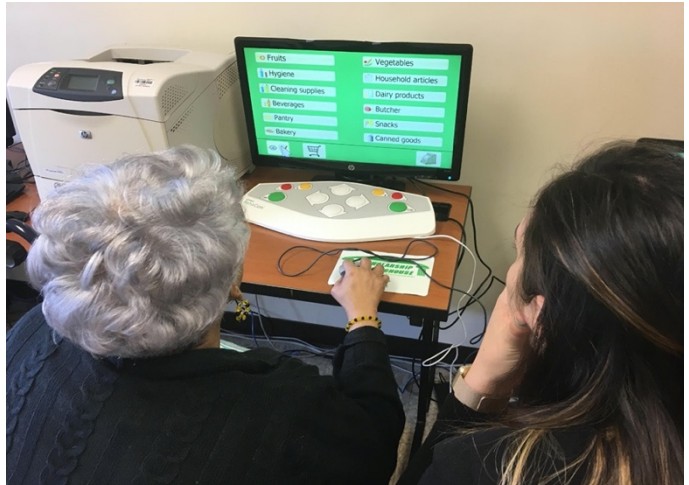 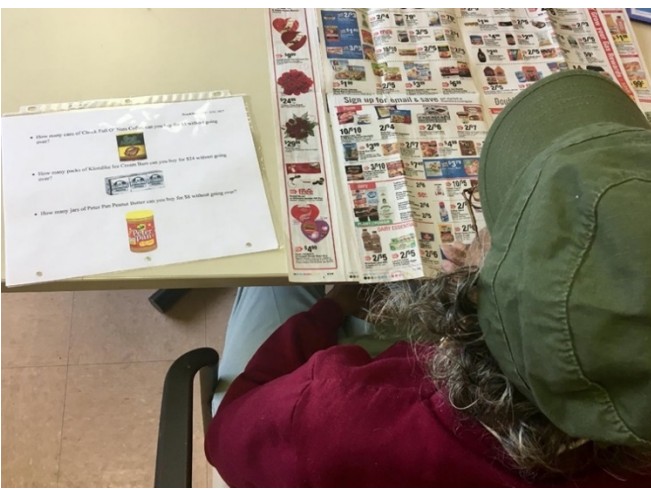

**Figure 2.** Example of the training: Comparison of a shopping task for both Computerized Cognitive Training (CCT) and Traditional Cognitive Training (TCT) groups.

The research assistants and the participants in the TCT group utilized a variety of paper-and-pencil activities within available and accessible rooms at the center. The research team designed the TCT participant-selected activities to be similar to the activity-based cognitive tasks within the modules of RehaCom. For example, to address occupation-based executive functioning in the TCT group, participants completed paper-based activities such as simulated shopping using a grocery store advertisement, including locating products in a flyer and calculating the total purchase cost (see Figure 2). Other TCT activities included crosswords, map reading, Sudoku, reading comprehension, and word games. They were inherently different from the RehaCom program due to the self-selection of the activities by the participants in both CCT and TCT groups.

*2.5. Data Analysis*

To contribute to the design, quantitative content analysis from open-ended questions used a conceptual analysis framework as suggested by Schreier for classifying the content [33]. Detailed participant responses for in-depth content analyses were lacking, so the open-ended survey was reviewed by two separate, trained reviewers (trained by the first author). Training consisted of independent analysis of a sample and comparison to determine interrater consistency. The reviewers categorized the content of the responses as positive or negative and quantified the frequency of occurrence of the positive or negative codes. A third trained reviewer assessed disagreements and confirmed results. Data were transformed into positive and negative responses to complete quantitative analysis.

McNemar's tests determined the significance of positive responses to perceptions between the two groups for quantitative content analysis. SPSS was utilized to check for normality, which was violated (except for session time analysis). Therefore, nonparametric tests were used (SPSS Version 26.0; IBM Corp.; Armonk, NY, USA). Wilcoxon signed-rank tests and Mann–Whitney U tests were used to determine differences within and between the two groups for executive function measure changes and utilization of the independent

*t*-test for session time differences between the two groups. The significance level was set at 0.05.

## 3. Results

The 20 eligible participants were randomized to either the CCT group (*n* = 10) or the TCT group (*n* = 10). Six participants (three in each group) did not complete the total 480 min of the training due to scheduling conflicts. They were excluded from the analysis, leaving seven participants in each group for data analysis. Characteristic data are included in Table 1. The Mann–Whitney U test (or Fisher's Exact Test for categorical data) was performed to compare baseline demographics between the two groups, with no significant differences at baseline.

**Table 1.** Demographic comparisons between groups.

| Demographic Characteristic | CCT (*n* = 7) M (SD) or *n* (%) | TCT (*n* = 7) M (SD) or *n* (%) | Fisher's Exact Test or Mann–Whitney U *p* |
|---|---|---|---|
| Sex (Female) | 4 (57.14%) | 6 (85.71%) | 0.56 |
| (Male) | 3 (42.86%) | 1 (14.29%) | |
| Age, Years | 68.00 (5.39) | 75.71 (10.00) | 0.21 |
| Race (Unreported) | 1 (14.29%) | 0 (0%) | 1.00 |
| (Black or African American) | 6 (85.71%) | 7 (100%) | |
| MoCA, baseline (x/30) | 22.29 (2.36) | 21.57 (3.26) | 0.51 |
| (Suspected MCI 17–25) | 7 (100%) | 6 (85.71%) | |
| Education (High school) | 5 (71.43%) | 5 (71.43%) | 1.00 |
| (≤12 years of education) | 2 (28.57%) | 2 (28.57%) | |
| Home assist (Lives alone) | 4 (57.14%) | 3 (42.86%) | 1.00 |
| (Lives with family) | 3 (42.86%) | 4 (57.14%) | |

Note: CCT = Computerized Cognitive Training, TCT = Traditional Cognitive Training, MoCA = Montreal Cognitive Assessment.

Table 2 describes the analyzed, coded, open-ended responses based on the seven questions of the survey. One participant in the TCT group declined to complete the open-ended survey. No significant differences existed in perspectives and preferences when comparing CCT to TCT. As questions were assessed regarding the continuation of the programs, there were no differences in perception of the two training techniques on helpfulness, enjoyment, perceived positivity, improvement suggestions, independence with training, and desire to continue with training. In addition, 100% of the 14 participants perceived the training as helpful and reported enjoyment of their training.

**Table 2.** Content analysis differences in preferences of the programming.

| Categorized Responses | CCT (*n* = 7) *n* (%) | TCT (*n* = 6) * *n* (%) | McNemar's Test *p* |
|---|---|---|---|
| 1. Helpful | 7 (100%) | 6 (100%) | 1.00 |
| 2. Enjoyment | 7 (100%) | 6 (100%) | 1.00 |
| 3. Opinion | 6 (85.7%) | 6 (100%) | 0.46 |
| 4. Independence to do on own | 3 (42.9%) | 3 (50%) | 1.00 |
| 5. Assistance support would be required to continue | 5 (71.4%) | 3 (50%) | 1.00 |
| 6. Continuation of training provided | 6 (85.7%) | 4 (66.7%) | 0.56 |

Note: Responses coded as "Yes" or "Positive." CCT = Computerized Cognitive Training, TCT = Traditional Cognitive Training. * One participant in the TCT group declined to complete the open-ended survey. The remaining six participants' responses from the TCT group were included.

The within-group cognitive performance changes are described in Table 3. On all measures, there were no significant differences from pretest to posttest time points in both the CCT and TCT groups. Similarly, there were no significant differences between the CCT and TCT groups for all cognitive measures using the Mann–Whitney U, including the EFPT

($p$ = 0.21), TMT-A ($p$ = 0.46), and TMT-B ($p$ = 0.26). As revealed by an independent *t*-test, there were no significant differences in preferences of the duration of the time of sessions (i.e., session tolerance) ($p$ = 0.81). The CCT average session time was 43.64 ($\pm$15.60) minutes, and TCT was 44.27 min ($\pm$16.10) (Table 4).

**Table 3.** Comparison of EFPT, TMT-A, and TMT-B changes within groups (*n* = 14).

| Cognitive Performance Outcome Measurement | CCT (*n* = 7) M (SD) | TCT (*n* = 7) M (SD) |
|---|---|---|
| EFPT Pre | 11.43 (4.83) | 25.29 (11.00) |
| Post | 11.14 (5.49) | 21.71 (9.69) |
| Pre-Post Change | −0.29 (4.57) | −3.57 (4.93) |
| *p* | 0.86 | 0.11 |
| TMT-A Pre | 66.57 (54.71) | 63.29 (18.39) |
| Post | 54.71 (27.26) | 66.29 (22.35) |
| Pre-Post Change | −11.86 (36.02) | 3 (13.87) |
| *p* | 0.61 | 0.61 |
| TMT-B Pre | 169.14 (187.00) | 176.71 (60.46) |
| Post | 118.14 (77.09) | 186.71 (57.67) |
| Pre-Post Change | −51 (133.62) | 10 (52.91) |
| *p* | 0.17 | 0.61 |

Note: CCT = Computerized Cognitive Training, TCT = Traditional Cognitive Training, EFPT = Executive Function Performance Test, TMT-A = Trail Making Test Part A, TMT-B = Trail Making Test Part B.

**Table 4.** Comparison of EFPT, TMT-A, and TMT-B changes, and the session preference time duration, comparisons within groups (*n* = 14).

| Cognitive Performance Outcome Measurement | CCT M (SD) | TCT M (SD) | *p* | | | |
|---|---|---|---|---|---|---|
| EFPT | −0.29 (4.57) | −3.57 (4.93) | 0.21 | | | |
| TMT-A (in seconds) | −11.86 (36.02) | 3 (13.87) | 0.46 | | | |
| TMT-B (in seconds) | −51 (133.62) | 10 (52.91) | 0.26 | | | |
| **Cognitive Performance Outcome Measurement** | **CCTM (SD)** | **TCT M (SD)** | ***p*** | ***t*** | ***df*** | **95% CI** |
| Duration of time of sessions (in minutes) | 43.64 (15.60) (total of 76 sessions among 7 subjects) | 44.27 (16.10) (total of 74 sessions among 7 subjects) | 0.81 | 0.24 | 12 | −4.45–5.71 |

Note: CCT = Computerized Cognitive Training, TCT = Traditional Cognitive Training, EFPT = Executive Function Performance Test, TMT-A = Trail Making Test Part A, TMT-B = Trail Making Test Part B.

## 4. Discussion

Our primary question was to determine the differences in how participants perceive CCT and TCT within a community-based, day programming environment for older adults. There are limited reports in the existing literature on older adults' preferences and tolerance of the length of time of cognitive training sessions within community settings when older adults are given the purview to control the timing of their sessions. The data from this study suggest that there are no significant differences in preferences of the timing duration of the training between CCT and TCT. Of the 76 sessions of CCT among seven participants and 74 sessions of TCT among seven participants, session averages were 43.64 ($\pm$15.60) minutes and 44.27 ($\pm$16.10) minutes, respectively, with similar standard deviations. Older adults may experience cognitive fatigue, regardless of the format of the cognitive content, which results from various factors related to aging, including age-related changes in energy

production [34]. The capacity to self-pace is a key ability to consider when monitoring the older adult during cognitively demanding activities, such as with the one-on-one supervision provided within our study. These findings are similar to Lampit et al.'s report that group-based CCT for at least 30 min is recommended [5]. The authors recommend 30 to 45 min of supervised cognitive training per session based on this sample. However, these results are preliminary with a small sample, and in future sessions, all participants should be closely monitored, perhaps via telehealth oversight from a trained practitioner.

Another objective of the study was to examine the preferences of type of training. All participants (except one participant who declined the survey) perceived the training as helpful and enjoyable. Although the literature suggests that CCT may be a preferred option to reduce costs of supervision while providing an equally effective alternative for TCT [17], 42.9% of CCT respondents and 50% of TCT respondents reported that they would feel comfortable continuing the training independently, i.e., without the assistance of a one-on-one trainer. This concept is not surprising, as socialization and companionship are reasons why participants attend the centers [4]. Additionally, evidence suggests that social engagement optimizes cognitive aging and social engagement is an activity associated with high cognitive function in older adults [1,35]. Socialization is a component of senior center programming and cognitive program planning and should have been formally measured to strengthen this point of evidence within the literature. In future examinations, socialization with a trainer or a built-in peer-to-peer component may be beneficial in the programming. From a programmatic decision-making standpoint, the survey responses suggest that the presence of a trainer, whether this role may be carried out by a trained student or an OT practitioner, should be continued from the members' perspectives to adhere to programming within a supportive environment. As care models are transforming, OT has a role in supporting cognitive aging further within technology-based models (e.g., telehealth), considering the efficacy within older adult care and facilitating the acceptance of technology [36,37].

*Limitations*

In addition to the limitation of the lack of a socialization measure, further limitations include that despite attempting to recruit a homogenous sample of older adults that may be experiencing normal cognitive aging, both groups were highly variable on the executive function measures, with large standard deviations at both pretest and posttest which indicates the presence of outliers (Table 3). Although the CCT groups improved in all three measurements of executive functioning, the significance and superiority cannot be determined due to variability and small sample size. The non-significant differences observed in the study could be due to the lack of power. Studies with moderate effect sizes of executive function outcomes used protocols of six months to two years [5]. However, this study was limited in time frame and scope to 12 weeks to measure the feasibility of both programs. Therefore, a longer and larger longitudinal study may result in more favorable outcomes for executive functioning and the consideration of follow-up questionnaires of self-initiation of the continuation of the training after study completion. Notably, due to the random assignment of participants, there was a 7-year age difference in the means of the two groups and large differences in EFPT mean scorings, and future studies should consider stratifying random assignment by age or cognitive performance. It should also be noted that the MoCA, which was used in recruitment, also includes a portion of the TMT-B, although it may not have influenced the results of the TMT-B used as an outcome measure. In addition, the study would have benefitted from several intake variables of questioning upon enrollment, such as previous experience with the use of computers, prior experience with cognitive training activities, and use of medications that may impact cognitive function. Future recruitment efforts should consider strategies to address the self-selection bias of enrollment in a study that focuses on the acceptance of technology.

Future study designs should consider a controlled treatment arm and the incorporation of a blinded researcher for instrument administration to reduce the potential for bias and

increase the study's overall validity. Future program planning should consider scheduling difficulties when working with independent, community-dwelling older adult participants while allowing flexibility for participant attendance.

We speculate that many community members declined participation because they were busy with the center's medical services and robust activity offerings. Although the measurable frequency and dosing duration among all subjects are ideal in a rigorous study design for a randomized control trial, a study design incorporated into participants' daily routine is ideal for the independent, diverse adult in a senior center environment. As daily routines and environments have altered with the onset of distancing, senior centers adjust programming needs to serve members who are not attending in-person due to capacity guidelines and other precautionary restrictions [18,38]. As rehabilitation activities may be integrated into telehealth delivery, the OT practitioner may work through the barriers of accessing technology within the client's home environment to continue to access cognitive programming. As commercially available cognitive training is available on app platforms for widely accessible use, the OT practitioner may utilize telehealth platforms to make recommendations to senior center clients. Recommendations may address the duration of session time to complete activities, determining the need for training support, and integrating socialization during the tasks. Further investigation on the programming feasibly is warranted within telehealth models. Within this study, the small sample of primarily Black participants was open to utilizing technology for cognitive training and valued the supportive component of a person who was available to assist.

## 5. Conclusions

Therapeutic practitioners are qualified to design and consult on the current existing programming systems, specifically to maximize cognitive well-being inclusive of diverse populations of older adults in the community. The inclusion of occupation-based CCT and TCT programming is feasible for diverse senior center programming, in alignment with evidence-based cognitive aging activity recommendations, if participants demonstrate the capacity for self-pacing for 30–45 min sessions. However, short-term improvements in executive functioning should not be expected but are worthy of longer-term observation. A socialization component would also be beneficial to measure for future studies and consider a telehealth component with one-on-one support for greater and safe access for all.

**Author Contributions:** Conceptualization, S.B., K.O. and D.S.; methodology, S.B., K.O. and D.S.; software, S.B.; validation, S.B. and A.M.P.; formal analysis, S.B.; investigation, S.B., K.O. and D.S.; resources, S.B. and A.M.P.; data curation, S.B.; writing—original draft preparation, S.B., K.O. and D.S.; writing—review and editing, S.B. and A.M.P.; visualization, S.B. and A.M.P.; supervision, S.B.; project administration, S.B., K.O. and D.S.; funding acquisition, S.B. All authors have read and agreed to the published version of the manuscript.

**Funding:** This research was funded by the Genesis CARES grant.

**Institutional Review Board Statement:** The study was conducted in accordance with the Declaration of Helsinki, and approved by the Institutional Review Board of the University of the Sciences in Philadelphia, project number 947324.

**Informed Consent Statement:** Informed consent was obtained from all subjects involved in the study.

**Data Availability Statement:** Not applicable.

**Acknowledgments:** Thank you to the Mercy Life-West Philadelphia members and staff for their support and participation. Thank you to Brianna Milstrey for her editing assistance.

**Conflicts of Interest:** The authors declare no conflict of interest. The funders had no role in the design of the study; in the collection, analyses, or interpretation of data; in the writing of the manuscript, or in the decision to publish the results.

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
