# Peer review of "A Feasibility Study of Two Cognitive Training Programs for Urban Community-Dwelling Older Adults"

_2673-9259, doi:10.3390/jal2020007_

Round 1
Reviewer 1 Report
This study compared two cognitive training programs – a traditional paper/pencil one and a commercially available computerized neurorehabilitation one – in two groups of mostly black (13/14) seniors (older than 54 years). Both programs were enjoyed by the participants, but they both had no effect on executive functioning as assessed by the EFPT and the TMT. Although the study has its merits as the results have practical relevance there are a number of methodical issues that need to be addressed by the authors:
Title: it should be mentioned that this is a feasibility trial
Abstract: There is no description of the participants while the title stresses the diverse nature of the cohort (BTW: can a sample be called diverse when 13 out of 14 are Black Americans?)
Methods: Crucial information on the participants is missing: how many were suffering from mild cognitive impairment? As the MMSE minimum score was set to 20, it is rather likely that some had significant cognitive deficits. With the minimum age set at 55 one wonders whether some of the participants were still occupied? Was there no upper age limit? Did the participants have experience with the use of computers? The mean age between the two groups varied considerably (7 years), why didn’t the authors try to control for this? The same holds for the EFPT performance. And here is the major problem of the study: with its small n and large variance, it is almost impossible to detect group differences. Can the authors at least speculate why so many potential participants declined to take part in this study? What about selection bias? Did the authors assess how many participants continued the training on their own after the study had ended? From personal experience most short-term interventions have no long-term effects as most participants return to their routines.
Author Response
Although the study has its merits as the results have practical relevance there are a number of methodical issues that need to be addressed by the authors:
- Title: it should be mentioned that this is a feasibility trial
Response: Title has been adjusted to “A Feasibility Study of Two Cognitive Training Programs for Urban Community-Dwelling Older Adults.”
- Abstract: There is no description of the participants while the title stresses the diverse nature of the cohort (BTW: can a sample be called diverse when 13 out of 14 are Black Americans?)
Response: We added to the abstract, Line 16, “Fourteen older adults (n=13 Black) were recruited.”
- Methods: Crucial information on the participants is missing: how many were suffering from mild cognitive impairment? As the MMSE minimum score was set to 20, it is rather likely that some had significant cognitive deficits. With the minimum age set at 55 one wonders whether some of the participants were still occupied?
Response: Thank you for the suggestion, we added a row in Table 1 to describe that most of the participants did likely have cognitive impairment at baseline, 100% in the CCT group and 85.71% in the TCT group. All participants attended the community day center so it is unlikely that the participants were not working (if this is what you mean by “occupied?”), however, we are unsure as this was not a question on our demographic intake form. We also added on Line 121 the clarification on the chart review of the MMSE, which now reads as, “…Mini-Mental State Exam (MMSE) scores less than 20/30, used solely to exclude moderate-to-severe dementia for recruitment efforts…”
- Was there no upper age limit?
Response: Thank you for seeking clarification, we have adjusted the wording in Line 550 to read as, “…was at least 55 years of age with no age limit cutoff…”
- Did the participants have experience with the use of computers?
Response: We are unsure, however, have added this statement to the limitations in Line 1070, “In addition, the study would have benefitted from a number of intake variables of questioning upon enrollment, such as previous experience with the use of computers.”
- The mean age between the two groups varied considerably (7 years), why didn’t the authors try to control for this? The same holds for the EFPT performance.
Response: Yes, we have also noted that there was a considerable difference between the two groups in mean ages and the EFPT. In future studies, stratified assignments based on age would be helpful to control for this difference. We have added in the limitations on Line 1065, “Notably, due to the random assignment of participants, there was a 7-year age difference in the means of the two groups, as well as large differences in EFPT mean scorings, and future studies should consider stratifying random assignment by age or cognitive performance.
- And here is the major problem of the study: with its small n and large variance, it is almost impossible to detect group differences. Can the authors at least speculate why so many potential participants declined to take part in this study?
Response: We have expanded on our previous statement in Line 1077 to now include, “Future program planning to consider scheduling difficulties that may arise when working with independent, community-dwelling older adult participants while allowing flexibility for participant attendance. We speculate that many community members declined participation because they were busy with the center's medical services and robust activity offerings.”
- What about selection bias?
Response: We have adjusted Line 1073 to now read as “In addition, the study would have benefitted from a number of intake variables of questioning upon enrollment, such as previous experience with the use of computers, and consider self-selection bias of enrollment in a study that focuses on the acceptance of technology.”
- Did the authors assess how many participants continued the training on their own after the study had ended?
Response: We did not inquire with the participants on follow-up. We have added to the limitations on Line 1063 “…as well as the consideration of follow-up questionnaires of self-initiation of the continuation of the trainings after study completion.”
- From personal experience, most short-term interventions have no long-term effects as most participants return to their routines.
Response: Thank you for your comment which we will consider for further study planning.
Reviewer 2 Report
This is an interesting feasibility study comparing the differences between an occupation-based computerized cognitive training program and a traditional cognitive training program. The study is well-written and I find that the results are interpreted carefully. I only have a few concerns that should be addressed by the authors: First, as a feasibility study with very low sample size, it will be good for the author to indicate clearly in their title that it is a feasibility or pilot study. That will helpful for readers to understand the purpose of the study. Second, although it is already briefly mentioned, the authors should highlight in their limitation section that the sample size of the current study is very small and any non-significant difference observed in the study could be simply due to the lack of power. This should be elaborated in clearly Third, the inter-rater between the two reviewers/coders should be reported in the Method or Results section.Author Response
This is an interesting feasibility study comparing the differences between an occupation-based computerized cognitive training program and a traditional cognitive training program. The study is well-written and I find that the results are interpreted carefully. I only have a few concerns that should be addressed by the authors:
- First, as a feasibility study with very low sample size, it will be good for the author to indicate clearly in their title that it is a feasibility or pilot study. That will help readers to understand the purpose of the study.
Response: Thank you, “Feasibility” has been added to the title.
- Second, although it is already briefly mentioned, the authors should highlight in their limitation section that the sample size of the current study is very small and any non-significant difference observed in the study could be simply due to the lack of power. This should be elaborated in clearly.
Response: Thank you for your suggested phrasing, we have added it to Line 1057 to read, “… and any non-significant differences observed in the study could be due to the lack of power.”
- Third, the inter-rater between the two reviewers/coders should be reported in the Method or Results section.
Response: Thank you, we have added on Line 668, “Training consisted of independent analysis of a sample and comparison to determine interrater consistency. The reviewers categorized the content….”
Reviewer 3 Report
Thank you for possibility to review this paper. The idea of this conducting this study is well-motivated and socially desired as assessment and supporting of normal cognitive ageing (with cognitive trainings) is a challenge for psychologists, occupational therapists and generally all professionals who work with elderly people. Article has many merits. Introduction is a good and synthetic base for the discussed topic and contains relevant information. However, I think Author/s could justify more the main objective of their study. Moreover, did this study obtain approval of commision for ethical standards? In limitation section could Authors add more information concerning other variables which could be analyzed in future research.
Author Response
Thank you for possibility to review this paper. The idea of this conducting this study is well-motivated and socially desired as assessment and supporting of normal cognitive ageing (with cognitive trainings) is a challenge for psychologists, occupational therapists and generally all professionals who work with elderly people. Article has many merits. Introduction is a good and synthetic base for the discussed topic and contains relevant information.
- However, I think Author/s could justify more the main objective of their study.
Response: Thank you, we added a closing sentence of the Background section on Line 363 to read as, “ There is limited evidence on the perception and outcomes of the different types of cognitive programming in community settings serving older adults, especially those from urban and diverse backgrounds. This study aims to inform rehabilitation professionals on the acceptance of cognitive programming for diverse populations and the feasibility of providing such programming in community.-based settings.”
- Moreover, did this study obtain approval of the commission for ethical standards?
Response: Thank you, although this information is on Line 1285 of the IRB study number, we have added the institution to Line 377.
- In limitation section could Authors add more information concerning other variables which could be analyzed in future research.
Response: Socialization and computer experience were added to the limitations.
Round 2
Reviewer 2 Report
The authors have sufficiently addressed all my comments. I appreciate their effort.